# Working with Young People at Risk of Suicidal Behaviour and Self-Harm: A Qualitative Study of Australian General Practitioners’ Perspectives

**DOI:** 10.3390/ijerph182412926

**Published:** 2021-12-08

**Authors:** India Bellairs-Walsh, Sadhbh J. Byrne, Sarah Bendall, Yael Perry, Karolina Krysinska, Ashleigh Lin, Maria Michail, Michelle Lamblin, Tina Yutong Li, Sarah Hetrick, Jo Robinson

**Affiliations:** 1Orygen, Parkville, VIC 3052, Australia; byrnes83@tcd.ie (S.J.B.); sarah.bendall@orygen.org.au (S.B.); karolina.krysinska@unimelb.edu.au (K.K.); michelle.lamblin@orygen.org.au (M.L.); tinayutong.li@health.qld.gov.au (T.Y.L.); jo.robinson@orygen.org.au (J.R.); 2Centre for Youth Mental Health, The University of Melbourne, Parkville, VIC 3010, Australia; s.hetrick@auckland.ac.nz; 3Centre for Global Health, Trinity College Dublin, D02 PN40 Dublin, Ireland; 4Telethon Kids Institute, Perth Children’s Hospital, Nedlands, WA 6009, Australia; yael.perry@telethonkids.org.au (Y.P.); ashleigh.lin@telethonkids.org.au (A.L.); 5Centre for Mental Health, School of Population and Global Health, The University of Melbourne, Parkville, VIC 3010, Australia; 6School of Psychology, Institute for Mental Health, University of Birmingham, Birmingham B15 2TT, UK; m.michail@bham.ac.uk; 7Townsville University Hospital, Douglas, QLD 4814, Australia; 8Department of Psychological Medicine, Faculty of Medical and Health Sciences, The University of Auckland, Auckland 1010, New Zealand

**Keywords:** suicide, suicidal behaviour, self-harm, primary care, general practitioners, young people, risk assessment, qualitative

## Abstract

General Practitioners (GPs) play a crucial role in the identification and support of young people at risk of suicidal behaviour and self-harm; however, no studies have explored GPs’ perspectives, approaches, challenges, and resource needs when working with this cohort in an Australian setting. This was a qualitative study where fifteen GPs (*M*_age_ = 45.25 years) from multiple clinics in Western Australia took part in semi-structured interviews, and data were analysed thematically. Seven main themes were identified: (1) working with young people has its unique challenges; (2) screening and assessment tools can help to manage uncertainty and discomfort; (3) going beyond tools–the dialogue and relationship are most important; (4) there are limits to what we can offer in the time available; (5) the service access and referral pathways lack clarity and coordination; (6) the provision of mental health support should not fall on GPs alone; and (7) more comprehensive training in suicide and self-harm is needed. The findings highlight a number of opportunities to enhance care and better assist GPs working with young people who present with suicidal behaviour and self-harm, including considerations for conducting assessments, targeted resources such as training, and system and service improvements.

## 1. Introduction

Suicide is the leading cause of death among young Australians, accounting for over one-third of deaths in people aged 15–24 years in 2020, and representing the highest number of years of potential life lost of any mortality cause [1,2]. Suicidal behaviours such as suicidal ideation, plans, attempts, and self-harm (non-fatal intentional self-harm with varying motives including the wish to die) [3] are more common, are key risk factors for future suicide, and are problematic in their own right [4,5,6,7,8,9]. 

Psychological problems are the most prevalent health issue managed by General Practitioners (GPs) in Australia, representing 70% of presenting issues in 2021 [10]. GP contact with young people with suicidal behaviours and self-harm is also common, and research has shown that the number of GP visits increases significantly before suicide in young people [11]. In one study, 40% of primary care practitioners reported treating a young person who had attempted suicide, and close to 70% saw a young person with suicidal ideation over one year [12]. Similarly, on average, 62% of people aged under 35 had contacted a GP in the year prior to death by suicide, and 23% had done so in the month prior [13,14]. Additionally, 58% of young people who had self-harmed had also seen their GP in the previous six months [15]. GPs are not only a frequent point of contact for young people experiencing mental health difficulties, they also act as a gateway to specialist services [16,17]. As such, GPs and the primary care setting represent a critical opportunity to identify and respond to suicidal behaviours and self-harm in young people [18]. Indeed, evidence suggests that significant reductions in Australia’s youth suicide rate could be achieved through improved detection and management of suicidal behaviour in primary care [19]. 

However, several studies from the UK and elsewhere have identified multiple systemic barriers to the provision of optimal care from GPs for this cohort. These include time constraints [20,21], heavy workloads [20], system resourcing issues with specialist mental health services such as long wait lists and a lack of integration and coordination [17,20,21], and an absence of specific mental health training on suicidal behaviours and self-harm [17,20,21,22,23]. As a result, GPs have reported problems with knowledge, confidence, and skills when managing suicidal behaviour and self-harm in young people [17,20,21,22,23,24], in particular the recognition of risk indicators, appropriate communication strategies for facilitating disclosure and engagement [20,22], and conducting risk assessments [17,22]. Recent evidence has outlined young people’s preferences for the way GPs work with young people at risk of suicidal behaviour and self-harm, and how they approach and conduct suicide and self-harm assessments [25], which are consistent with current recommendations to undertake psychosocial-based assessments that are collaborative and client-centred [26,27,28,29]. However, little research has focused on the views and needs of Australian GPs themselves when working with suicidal behaviour and self-harm in young people, nor explored whether GPs’ approaches are consistent with young people’s preferences. 

Taken together, it is important to fully understand the needs of GPs when supporting this population, including any challenges that may exist. Although a large proportion of the existing research on GPs’ experiences of working with suicide and self-harm presentations in young people comes from international sources, particularly the UK [20,21,24], to date, no studies have examined GPs’ perspectives, approaches, challenges, and resource needs in an Australian setting. Thus, there is a need for Australian-based research that examines current practice to inform the development of targeted resources to assist GPs in the delivery of optimal care. Furthermore, this research also aims to add to the growing literature base on GPs’ experiences of youth suicide and self-harm internationally. Specifically, the research question was: *what are the views and experiences of GPs regarding the identification, assessment, and care of young people who present at risk of suicidal behaviour and self-harm?*

## 2. Materials and Methods

The methodological details of the study are presented in accordance with the Consolidated Criteria for Reporting Qualitative Research (COREQ) [30], with the COREQ checklist provided in Appendix A.

### 2.1. Study Design

This was a qualitative study that used a combination of face-to-face individual and group interviews with GPs. Although individual and group interviews are recognised to elicit somewhat different data, the two methods can provide complementary perspectives on a topic, and can improve data richness [31,32]. The theoretical framework and orientation informing data collection and analysis was an essentialist/realist, experiential, inductive approach, where interviews and data analysis focused on representing participants’ expressed realities as told [33]. 

### 2.2. Setting, Recruitment, and Sample

The study was conducted across June and November 2018, in the Perth and Perth South Primary Health Network (PHN) regions of Western Australia by researchers from Orygen in Melbourne and the Telethon Kids Institute in Perth, under the auspices of the National Suicide Prevention Trial [34]. The Perth South region was selected by the Australian Government as one of 12 National Suicide Prevention Trial Sites, with a priority population focus on young people, due to an identified higher rate of suicide in this area over an extended period [35]. Additionally, the region includes some of Australia’s most disadvantaged areas [36].

GPs were purposively sampled via their clinics, with seven different clinics in the region selected to ensure a diverse range of types, sizes, and locations, including large free or low-cost general practice clinics, youth or young adult-targeted medical clinics, and specialised youth mental health services that provided additional primary care support. Clinics were identified and contacted directly about the study with assistance from the WA Primary Health Alliance (WAPHA) (which operates Western Australia’s PHNs), and interested GPs were invited to take part. Recruitment ceased when the data were considered to address the guidelines of information power—that is, data were sufficiently rich and novel to address the aim of the study, to support the analysis, and to generate new understandings [37].

Fifteen GPs (nine males, six females) took part in the study. Demographic information was provided by eight participants; the other seven did not complete the form. Of these, the mean age was 45.25 years (*SD* = 6.45; range: 37–53). English was the primary language spoken at home for six of the eight GPs, and none identified as being Aboriginal and/or as Torres Strait Islander. Four GPs practiced primarily in suburban locations, two in metropolitan locations, one in both suburban and metropolitan locations, and one in a regional location. The average duration of professional practice for participants was 12.63 years (*SD* = 10.14; range: 2–30).

### 2.3. Materials

A brief questionnaire was used to collect demographic information, and a semi-structured, open-ended interview schedule was used to facilitate the interviews. The schedule was informed by the existing literature, and the identification of research gaps and important questions to ask GPs. This covered the following six key topics: (1) previous experiences supporting young people who present with suicidal behaviour; (2) identifying suicidal behaviour and self-harm; (3) current assessment and management practices; (4) challenges when supporting young people with suicidal behaviour and self-harm; (5) perspectives on what constitutes best-practice, and best-practice implementation barriers; and (6) practitioner training and resource gaps, needs, and preferences. Depression, as a known risk factor for both suicidal behaviour and self-harm [7,38], was also included in the interview question wording. However, the reported experiences of GPs, and thus the paper, takes a much wider focus of suicidal behaviour and self-harm—one that is not solely tied to depressive symptomology or a depression diagnosis. We wanted to capture the broader, and potentially trans-diagnostic instances of suicidal behaviour and self-harm, in addition to those that occur outside of a mental health diagnosis. The interview schedule is outlined in Appendix A.

### 2.4. Procedures

Participating GPs were provided with the study information and given the choice of taking part in either an individual or group interview. In total, there were five individual interviews (quote identifiers reported as P1, P2, etc.) and three group interviews; with three participants in the first (reported as GI 1), four in the second (GI 2), and three in the third (GI 3). As there were multiple participants in the group interviews, specific participant identifiers for these could not be determined from the transcripts and audio recordings. Identifiers for the group interviews are thus instead reported by gender, with an additional identifier number reported wherever possible. Interviews were primarily facilitated by author J.R., with assistance from Y.P., S.J.B., and A.L., and conducted at the GPs’ clinics. All interview facilitators were researchers with psychology-related backgrounds and qualifications, who had previous experience in qualitative interviewing, and a range of professional research experience in healthcare systems with both practitioners and young people, and on the topic of suicide and self-harm. Interviews ranged from 15 to 53 min in length (*M* = 37 min) and were audio-recorded, then transcribed by a professional transcription service.

### 2.5. Data Analysis

Data were imported into the qualitative management software NVivo 12 [39] to assist with analysis. To analyse the data, author I.B.W. used the reflexive thematic analysis approach described by Braun and Clarke [33,40,41,42], involving the steps of data familiarisation, generating initial codes, and developing overarching themes and relationships. Thematic maps were constructed, and memo-writing was undertaken throughout the analysis. Following the initial theme development stages, I.B.W. reviewed and refined the thematic structure with assistance from S.J.B., K.K., and S.B., to establish a concise and coherent thematic account. This also supported analysis fidelity by including multiple judges and perspectives throughout the analysis process, which sought to improve consensus about the meaning and interpretation of the data and a representative thematic structure [43]. Disconfirming case analysis was also conducted throughout to account for data that was discrepant to the themes and patterns identified [44]. Any differing perspectives identified within the themes are reported below. 

## 3. Results

The analysis generated seven main themes, which were as follows: Working with young people has its unique challenges;Screening and assessment tools can help to manage uncertainty and discomfort;Going beyond tools—the dialogue and relationship are most important;There are limits to what we can offer in the time available;The service access and referral pathways lack clarity and coordination;The provision of mental health support should not fall on GPs alone; andMore comprehensive training in suicide and self-harm is needed.

### 3.1. Working with Young People Has Its Unique Challenges

A number of challenges unique to working with young people were identified. One such challenge was determining the risk of suicidal behaviour in young people, which was attributed to the complexities of this developmental stage. Establishing whether a young person was just dealing with *“teenage angst”*, or whether they were experiencing a *“real mental health problem”* with true suicidal intent, could be challenging: *“It’s more difficult to, I suppose, gauge the risk… [for] teenagers it’s such a difficult time, and there’s a lot of emotional upheaval”* (Male 1, GI 1). This made it *“really difficult to assess whether they are going to do something or not”* (Male, GI 2). Participants discussed how predicting suicidal behaviour in young people was complicated due to *“their lack of ability to make rational decisions”*, and how young people’s behaviour could be *“a lot more impulsive”* and *“chaotic”* (Male 1, Male 2, GI 1). One GP believed that warning signs of suicide or self-harm were more difficult to identify in young people or were often absent altogether: *“Most of the time they don’t give any clue [to] what they are planning for”* (Female, GI 1).

Some participants experienced communication barriers with young people; feeling that communication was *“more difficult with the teenagers as compared to the adult”* (Male, GI 2). When the young person’s stated presenting concern was not mental health or suicide/self-harm-related, it was particularly hard to raise these topics with them: 


*“How do we put it to ask about that… because they don’t feel comfortable, but if they already know about the situation and they’re just coming for help then that’s totally different”*
(Female, GI 2).

GPs noted that young people could experience difficulties disclosing suicidal ideation or self-harm, further impacting their ability to identify risk. They described how young people might feel *“shyness”* (Male, GI 2), acknowledging that they would *“have to be pretty brave to go in and say, ‘I’m suicidal’”* (P4). 

Another challenge specific to young people was the involvement of parents. Parental attendance at the GP appointment when a younger adolescent was engaging in self-harm or was suicidal was a common experience. Participants felt that young people had a strong need for privacy and confidentiality, particularly around parental and family involvement: *“[We need to be] very cautious and sensitive to their confidentiality”* (Male, GI 2). Although parent involvement had many positives, such as being a source of *“collateral history”* (Male, GI 3), a *“support person”* (P2), and helping to monitor the young person’s safety, it was something GPs had to navigate carefully. Some young people could come from *“dysfunctional families”* with *“no family support”* (Male, GI 2), and others *“didn’t want [their] parents involved”* (P4). This created difficulties for the GP, as the attendance of parents at the appointment could interfere with the young person’s disclosure and the GP’s subsequent assessment: *“Sometimes they’ve been brought in by their parents and the parent talks for them, and sometimes [the] patient doesn’t even want to be here”* (Male, GI 2). Consequently, negotiating confidentiality when a young person was deemed to be at high risk of suicide or engaging in self-harm was seen as a challenging, but necessary, process to supporting their safety: *“Obviously you’ve got confidentiality, but then obviously if you’ve got someone who’s actually suicidal, then you know… you’ve got to keep them safe”* (P1).

### 3.2. Screening and Assessment Tools Can Help to Manage Uncertainty and Discomfort

Several participants described how the process of working with suicidal or self-harm presentations in young people could create feelings of discomfort, worry, and uncertainty. The lack of objectivity in regards to assessing mental health compared to physical health issues was seen as a barrier to accurately determining a young people’s safety: *“Mental health is really hard to protocolise. With physical health you’ve got blood parameters…there’s a lot more objectivity about it”* (Male, GI 1). Instead, GPs often needed to *“rely on a sense”* and *“experience and intuition”* (Male, GI 1) when assessing risk, in the absence of clear and definite indicators. Participants also questioned whether GPs would *“be prepared to just ask the question [about suicidality/self-harm]”* (P4) and highlighted how GPs might be reluctant to initiate this conversation *“because it’s awkward and difficult”* (P2). 

Participants expressed a range of views on the use of tools and assessments to identify, assess, and manage suicidal behaviour and self-harm in young people. They tended to use general screening instruments or *“tick-box questionnaires”* (P3) to initially identify potential risk factors for suicide in young people; most commonly the Depression Anxiety Stress Scales—Short Form [45,46] or the Kessler Psychological Stress Scale [47]. The HEEADSSS [48,49] assessment was also described as a general screener for psychosocial risk factors and the potential presence of suicidality/self-harm, which was recommended *“through the college [The Royal Australian College of General Practitioners (RACGP)]”* (Male, GI 2). However, several participants also identified that there were no *“set protocols”* (Male, GI 1) in place at their clinics to identify and respond to suicidal behaviour, and some wanted a more standardised protocol *“so that everyone knows what to do”* (Male 1, GI 3). They felt that risk assessment standardisation and documentation would help to protect the patient’s safety, and that of the GP’s, from an adverse event by providing medico-legal evidence. However, one GP was particularly critical of using risk assessment tools and protocols to alleviate GPs’ fears, and felt that these approaches demonstrated a lack of patient care: 


*“Most of the times it’s actually about the GP’s risk, the way that they’re going to be trawled through a Medical Board or they’re worried that they’re going to get sued. They’re not actually genuinely worried about the patient”*
(P5).

Similarly, some interviewees raised concerns that standardised risk assessments lacked evidence for prediction and accuracy and therefore had limited value: *“The last time I looked at the evidence for actual formal risk assessments… they were not actually that flash with predicting whether or not someone could complete suicide”* (P2). Screening and assessment tools could also be *“impediments to the therapeutic relationship”* (P5). Despite these concerns, several participants expressed a desire for tools or guides to help them to conduct assessments and better conceptualise suicide and self-harm risk. They highlighted the value of *“knowing your certain questions you’re going to ask”* (P4), with simplified and brief guides viewed as useful, that could provide specific example questions, help to differentiate the level or immediacy of risk: *“Mild, moderate, severe”* (P2), and recommend corresponding management strategies and referral pathways: 


*“Like a flow chart to say, ‘Person at risk, but not acutely at risk, needs help with this, then refer them to headspace, refer them to da-da-da’… with maybe more specific questions to ask, that might give us three or four questions to try and assess risk a bit more”*
(P1).

Participants also felt that risk assessment tools, checklists, and frameworks could be valuable when used as a prompt: *“So that I don’t forget something”* (P2), or to assist those with limited experience. As GPs’ experience grew, their need for such tools may decrease, as they would *“[get] a feel for who is at risk and who isn’t”* (Male, GI 1). However, regardless of the tools or measures used, ambiguity, uncertainty, and discomfort were seen to be an inevitable part of the suicide and self-harm assessment process, as no tool or intervention could ever truly guarantee a young person’s safety: 


*“It’s always going to put you on edge when somebody talks about suicide… that goes with it whether you’ve done a piece of paper, or whether you’re talking to somebody. Whether you send them off with a safety plan, or whether you haven’t. Whether you’ve sent them off with numbers that they can contact or said to them, ‘Go to the Emergency Department if things escalate’. There’s always going to be that level of discomfort. Have I actually handled things ideally? Is there still a risk? Is this patient going to be safe? Could things escalate? Am I going to get a call?”*
(P3).

### 3.3. Going beyond Tools—The Dialogue and Relationship Are Most Important

Despite some participants highlighting the benefits to screening and “tick-box”-type tools to help assess for the presence of suicidal behaviour and self-harm, they outlined that assessment tools should not be used rigidly at the expense of recognising and supporting a young person’s needs: *“If something comes up, follow that. Don’t just follow the framework… also follow the young person”* (P2). Participants suggested that any assessment tools needed to be flexible and applied in a conversational, adaptable manner: *“Just weave it into your conversation… if you could actually teach GPs that you don’t have to go ‘A, B, C, D, E, F, G’*” (P5). Most importantly, having a collaborative dialogue with a young person, over and above the use of any tools or instruments, was viewed as key to the assessment process: 


*“It’s more important to be able to have a dialogue with your patient and some direct discussion about what’s going on in their head, what are they thinking about”*
(P3).

The content of conversations and dialogue with young people also aimed to assess the level and immediacy of potential suicide risk. Identifying the frequency and intensity of ideation, plans for self-harm, the presence of risk and *“protective factors”* (P3), and looking for changes relating to hopelessness, worthlessness, helplessness, or a loss of interest or motivation were strategies used by GPs. One participant acknowledged that although GPs could initially feel confronted by disclosure of suicidality/self-harm, it was important to have a thorough conversation with the young person, to gain an in-depth understanding of their experiences and to identify the most appropriate treatment pathway: 


*“It is hard when you haven’t worked in this field, to separate the sheep from the goats sometimes. If a young person tells you, ‘Yep, I have suicidal thoughts and I have them every day,’ for that GP it’s like, ‘Wow’. But then when you investigate further, actually they’ve had them every day for three years. And actually, they haven’t acted on them in that time. And actually, they’ve got chronic emotional dysregulation difficulties. So, firing them off to ED in that instance might not be the most appropriate thing, might it?”*
(P2).

Remaining vigilant to any situational factors in the young person’s life that may contribute to suicide risk, rather than just relying on the presence of diagnostic or mental health labels, was also important: 


*“Pretty much any situation [I would ask about risk]… to me it doesn’t need a specific mental health title. Any young person who’s experiencing emotional distress can have suicidal thoughts”*
(P2).

Underpinning all conversations or assessments about suicide or self-harm was the importance of having a positive relationship. Some participants felt that the type of relationship young people wanted from GPs was primarily therapeutic, where being listened to and having a connection were highly valued: *“They want to be connected with. They don’t want ‘process me, process me, process me’… [Young people] just want to be heard”* (P5). Participants emphasised how *“being able to vent… then [the GP] being able to say, ‘I hear you’”* could *“do a huge amount”* (P3) in supporting a young person’s distress, and how the development of a good relationship with a young person was a critical component of care that could influence ongoing engagement: *“If someone doesn’t have good rapport… why are they going to come back and see them?”* (P4).

GPs were very cognisant that the relationship they had with the young person could impact the likelihood of disclosure. Being attentive to the young person’s needs for privacy was an approach used to facilitate a trusting relationship: *“I tend to ask about risk when Mum or Dad has stepped out of the room”* (P2). Discussing confidentiality with the young person from the beginning, and working collaboratively with them to identify the most appropriate source of support, was also important. When a young person did disclose self-harm behaviour, taking a non-judgemental stance also allowed exploration of the behaviour’s therapeutic function: 


*“I’d think why are they doing that and what’s that release of emotion? I wouldn’t necessarily say to them, ‘This is something that you need to stop immediately’, because often that’s not the right thing to say to them I don’t think”*
(P1).

### 3.4. There Are Limits to What We Can Offer in the Time Available

Time constraints in the consultation were identified as a significant challenge for GPs. Participants described how this compelled them to be rapid when asking about suicidality: *“That’s why we’re probably more likely to be direct in some ways”* (P1), and how they were more likely to use *“tick-box assessment types of tools”* for efficiency (P3). Often participants grappled with a lack of time needed to develop an adequate rapport to successfully approach the subject of suicide and self-harm, and then to undertake a full, comprehensive assessment. This meant they often postponed these assessments to follow-up appointments: *“You don’t know if it’s going to be five minutes or 25 minutes or 35 minutes, so I’ll say, ‘Ok… I’ll see you tomorrow’”* (Male, GI 3). This was especially *“harder with a new patient”* (P1) where a rapport and patient history were lacking, or for an unexpected presentation of risk. The unpredictability of the time needed to address these situations and the anticipated delay on other patients was viewed as highly stressful: 


*“How can you possibly deal with a crisis situation in that period of time? It’s either rushed through and not managed… [or] the GP themselves are hugely stressed because of that workload and that knowledge that everybody else is still waiting as well”*
(P3).

Some participants expressed that GPs can be hesitant to ask about suicidal behaviour even if they did feel confident in how to phrase the conversation, as doing so could then create more responsibility and time pressure on the GP to then have to manage risk if it was identified: *“They might not want to open that can of worms”* (P2). One GP expressed that universal mental health and suicide screening questionnaires for young people could potentially alleviate some of the time barriers on GPs, as *“you can get more information in a timely fashion”*; however, they may also backfire and inadvertently cause more time pressure: 


*“[If] we look at the questionnaire and it looks as though they’ve had suicidal thoughts, okay, it’s probably not that serious, but they’ve ticked the box. You have to do your full assessment… the GP cannot manage that in the consultation time”*
(P3).

Clinic structures, billing, and funding considerations were also seen to impact the amount of time GPs could offer, and thus the level of patient attention they could provide: 

*“If you’re fully bulk-billing* (providing free appointments supported through Australia’s universal public healthcare system), *then often it’s 10 minute appointments, it’s just churning through large numbers of patients… there’s no capacity to really spend time on mental health issues… with the current Government funding for GP consultations… they’re going to need to do numbers to get through it”*(Male, GI 1).

Despite time pressures and constraints, several participants described how, if they encountered a young person presenting with suicidal behaviour or self-harm, they would try to offer them as much time as they needed, even if *“that means you run late for the next one, that’s just what it takes”* (P1).

### 3.5. The Service Access and Referral Pathways Lack Clarity and Coordination

A lack of clear access, integration, and communication between services was highlighted as a key problem for GPs. When referrals to specialist services were needed, participants expressed frustration with a lack of information and signposting from external services intended to support young people, and poor communication regarding referral pathways: 

Female: *“We don’t know about many of them…”* Male 2: *“Everyone’s working in their own little silo, no one talks to each other… at the moment it’s like using the patient as a ping-pong, [they] send to you, [you] send them back… it’s really hard to actually communicate effectively between you”*(Female, Male 2, GI 1).

Dealing with *“complex”* and *“wishy-washy”* (P3) service entry criteria between primary care and specialist services left GPs feeling unsupported: *“The mental health services or anybody that assists… rarely make you feel as though they’re doing you an enormous help, when it’s actually supposed to be their job”* (P3). Participants described incongruent perspectives on what constituted the need for service involvement between themselves and specialist services, leading to young people being *“bounced back”*, and a feeling of *“hitting barriers when I’m trying to get people assessed”* (Male 1, GI 1). Such *“fragmented”* services (Male, GI 1) and perceived system failures created an overwhelming sense of frustration and powerlessness, where *“people [are] being pushed from pillar to post and nobody [is] taking responsibility”* (P5). Gaps in the healthcare system for supporting young people who presented with greater than low levels of risk were seen as problematic: *“There’s no sort of in-between for the intermediate, slightly higher risk than what we’d be comfortable managing on our own”* (Male, GI 1); *“There’s a lot of availability for more mild-to-moderate cases, but there is a huge gap in the provision of services for acute, urgent situations”* (P3). This meant that when risk was assessed as severe or *“imminent”*, participants often referred young people to the Emergency Department (ED). Despite knowing that this could be quite a difficult experience for a young person, and ED admission generally being *“something we’d avoid”* (P4) and only a *“short-term [solution]”* (Male, GI 2), the ED was frequently the only option available.

Many participants also wanted training and resources that signposted services and offered transparency around available referral pathways to support young people: *“Who’s available, what it costs, how long it takes to access those resources”* (P3). Additionally, having access to another professional who acted as a *“regular point of contact”* such as a *“GP liaison nurse”* or *“triage nurse”* who was *“aware of all the services”* available was seen as useful (Male 2, GI 1). Tertiary-level child and adolescent mental health services (e.g., Child and Adolescent Mental Health Services—CAMHS) were specifically mentioned as a service that GPs felt they needed better training from regarding referral and access criteria: 


*“What they offer would be useful … when it is appropriate to send to whom, and so we feel like we can direct… if we knew all the options then we could use those services much more appropriately”*
(Male 2, GI 1).

However, one participant emphasised that it was not necessarily enough to just provide education or simple directives such as a *“piece of paper”* that advised them to *“call up your local mental health service”* (P3). Instead, systemic changes were needed, where the GP could trust that if they did refer as advised, they would be met with cooperation and availability: *“[It needs to be] really directive to say, ‘There’s a memorandum of understanding, and they will respond’”* (P3).

### 3.6. The Provision of Mental Health Support Should Not Fall on GPs Alone

Participants expressed frustration at having to provide care that they felt was outside of their remit as a GP, and had concerns with expectations around the general management of mental health, suicide, and self-harm-related presentations in young people. Some participants felt that it wasn’t necessarily the GPs’ responsibility to specialise in these areas, and described how GPs often self-selected when seeing patients who presented with suicidality/self-harm: *“We seem to see a tendency that some doctors in the practice will say, ‘Look, that’s not what I’m good at. You’re best off seeing (a), (b), and (c)’ and so on”* (Male, GI 1). However, one participant emphasised that competence with such presentations should be an inherent, non-negotiable part of a GP’s role: 


*“I’ve had young people tell me that they saw a GP and they’ve said, ‘Oh, I don’t do mental health’… it’s like—then you shouldn’t be a GP, go and get another job… You should not be a GP without mental health”*
(P5).

Due to resourcing issues and complicated admission criteria for specialist services, some participants felt as if *“the increasing burden of mental health problems gets transferred to GPs”*, and that they were forced to take on *“the more pointy end of the mental health spectrum”*—often being expected to operate as an *“emergency service”* or as a *“de facto mental health worker”* (P3), which was felt to be outside of their skillset and capacity. Extensive waiting periods for psychiatrists and specialist services also contributed to participants feeling like they were *“[trying to] manage someone”* who was suicidal on their own (P1). 

Generally, participants perceived their role in the young person’s care as being referral agents or stepping-stones: *“Ultimately… getting that young person to where they need to go”* (P2). However, some questioned whether this approach of *“just information-gathering, assessing risk and then referring them onto someone else to deal with it”* (Male 2, GI 1) was actually helpful for young people. However, they described a sense of powerlessness in not being able to offer much more, due to time, skill, and system constraints: 


*“Maybe we downplay our role in it sometimes as just the kind of first point of call and then send them to someone who can really engage… because we do feel pretty powerless, we’re stuck in a room, we don’t have any idea of what their life actually is like out there for them… I don’t see them for any length of time, and we’re not trained in psychological interventions”*
(Male 2, GI 1).

To combat this, many participants strongly emphasised the need for support from mental health professionals to assist in caring for a young person with suicidal behaviour and self-harm. The provision of a multidisciplinary care team, ideally delivered in-house, was desired: 


*“A system where if you have something like this happen where a child is self-harming then… automatically a social worker would be reviewing the social aspect of things, you’d have a multidisciplinary team approach it. You’d get the psychologist and the psychiatry team as well, and we’d all be liaising”*
(Male 1, GI 1).

Several participants highlighted that although their clinic did have systems in place for accessing mental health support, these were quite limited and felt to be inadequate. For example, one clinic had access to a counsellor, but they were not youth-specific and their availability was infrequent. Mental health professionals who were able to *“spend more than 15 minutes with [the young person]”* (P4) were felt by participants to be better suited to undertaking a comprehensive suicide and self-harm assessment, providing any necessary therapeutic intervention, and conducting safety planning with young people:


*“Somebody to actually go through and create a safety plan with them and to show them the app and say, ‘This is how it works. Let’s fill it all in’… to do it properly you can’t do it really in one minute or say, ‘Oh here’s a safety plan, fill it in. Share it with somebody’. Maybe that’s better than nothing, but it’s not optimal”*
(P3).

### 3.7. More Comprehensive Training in Suicide and Self-Harm Is Needed

All participants indicated that they had some basic mental health training from a range of sources. These included rotations as part of their initial medical training, which some felt gave *“very little exposure”* to suicide and self-harm (Male 1, GI 2), in addition to previous employment, conferences, and workshops and modules through the RACGP or other GP training providers. Sometimes these included more specific training in suicide and self-harm assessment and management, however, this was rare. Although some participants felt reasonably competent in the area of suicide and self-harm, others indicated that they would benefit from more comprehensive training and that such topics should be *“part of the GP training curriculum… in medical school and GP training”* (P4). This was particularly important given that many participants reported mental health and suicide/self-harm-related presentations as being a common part of their daily practice, for some occurring up to *“two or three [times] each day”* (P2). Very few participants had received suicide and self-harm training that specifically focused on working with young people, despite reporting that these presentations were becoming increasingly frequent. 

Participants expressed preferences for content that covered risk factors for suicide, in addition to training in the use of safety plans, which was repeatedly cited as a way to help alleviate some of the discomfort and worry around managing risk: 


*“Safety plans. That’s one thing I sometimes struggle to formalise… you have a plan, a safety plan that you agree with the relative as well… formalising that is sometimes quite difficult, and to actually give them something to take away”*
(Male, GI 1).

They also felt that any suicide or self-harm-related training should include recognised points that contributed to any professional development requirements and that ideally, *“active learning”* components (P2) with opportunities for skill application, such as role-plays, would be included. There was enthusiasm for both face-to-face and online training, with online delivery methods preferred when their time was limited: 


*“Some of these online modules are good. I think it’s always the time pressure isn’t it… we can’t go to every single lunch or evening presentation”*
(P1).

A *“combination of the two”* modes of delivery (Female, GI 2), where the same training material could be delivered in both face-to-face and online settings, was endorsed. The topics of suicide and self-harm were seen to be complementary and could be combined into the same session. The involvement of both young people and GPs in developing or delivering suicide and self-harm training content was also seen as being beneficial. Having a GP educating other GPs, *“because they can answer the practical questions”*, and a young person who could *“[talk] about their stories”* would enhance GPs’ understanding, as *“we all learn more by stories”* (P5). Despite participants asserting that good rapport and communication were critical skills in the identification, assessment, and management of suicidal behaviour and self-harm, some expressed doubt about whether training in these areas would be effective. These participants felt that rapport was *“hard to teach”* (P4), and that a sense of genuine patient care was either fundamentally present or not: 


*“By the time you’re a qualified GP, I hate to say it, but I think your bedside manner and your communication skills are pretty much set… I don’t think any lecture or tutorial is really going to change that. You’re going to revert to instinct, whether you truly care about the patient in front of you”*
(Male 2, GI 1).

Similarly, GPs’ interest in the topic of suicide/self-harm could influence the impact and uptake of any training: *“If you don’t like [working with mental health/suicide/self-harm], it doesn’t matter how much training you have, you’re never going to be good at it, I’m afraid”* (P5).

## 4. Discussion

This study identified a number of inter-related challenges and complexities that Australian GPs experience when working with young people at risk of suicidal behaviour and self-harm; however, it also underlines the importance of delivering best-practice primary care. Many of these challenges were not unique to this study and are consistent with findings from the UK and elsewhere [20,21,22,23,24], and thus contribute more broadly to international evidence on this topic. Although some issues were unique to working with young people, broader system barriers may apply to other types of presentations and demographics GPs treat (for example, having enough time to address presenting concerns [50,51]). However, it is essential to acknowledge their impact, as GPs believed these had significant consequences on their ability to identify and respond to suicidal behaviour and self-harm appropriately, and that these contributed to more specific challenges, such as the development of a therapeutic relationship, the processes and approaches taken to conducting assessments, and the level of patient attention and management offered. Lastly, when comparing GPs’ perspectives on working with suicidal behaviour and self-harm with the preferences of young people themselves [25], we identified a number of both converging and conflicting views and needs between these groups, which are important to highlight.

### 4.1. Key Findings

Primary care plays an essential role in early intervention with young people and in preventing suicide and self-harm trajectories [11,18,52]. However, our study identified several barriers that may hinder early identification and management. Uncertainty in how to initially “ask the question” or frame the conversation about suicide or self-harm was highlighted. Additionally, even when GPs did feel confident, time constraints, fear of increasing workloads, and feelings of having limited support across the broader system if risk was identified, still led to hesitancy. Research shows that simply enquiring about the presence of suicidal thoughts and having a strong therapeutic interaction with the GP can help to alleviate suicidal distress [53,54,55,56]. Although GPs recognised how valuable these approaches were, some felt limited in their ability to achieve this. Overarching funding structures for primary care also created pressure to turn over patients quickly, at a detriment to the therapeutic relationship. Young people have previously expressed feeling “treated like a number” and are pushed “in and out” of GP consultations [25], which could be attributed to this system constraint. However, it is more common for younger people to attend different GPs or GP clinics [57,58], which may also impact on the development of an ongoing therapeutic relationship. 

GPs also had conflicting views on their roles and responsibilities for the care of suicidal behaviour and self-harm in young people. Similar to past findings [21], GPs perceived their role as encompassing short-term management that involved brief assessment and referral. However, they were reluctant to take on what they perceived to be more intensive or complex management, which was partly influenced by service integration issues and the resulting pressure on GPs. For some, conducting comprehensive suicide and self-harm assessments and providing (what are often considered to be) brief psychosocial interventions such as safety planning were viewed as the responsibility of mental health professionals, and outside the scope of primary care. This demonstrates discrepancies between what young people want (i.e., safety planning plus a comprehensive and holistic exploration of risk [25]) and what some GPs perceive to be feasible. It is also inconsistent with recommendations that psychological care should not be the sole responsibility of mental health professionals, and that primary care plays an important role in mental health intervention [59]. Taken together, these reported barriers are problematic, given that if GPs postpone appointments without conducting a comprehensive assessment, refer young people elsewhere for simple interventions, or even avoid enquiring about suicide and self-harm altogether, disengagement is likely.

However, there were also positives in the ways GPs approached working with young people. Dealing with hesitancy from young people to disclose suicidality and self-harm, in addition to balancing family attendance at appointments and mediating confidentiality, have been identified previously as challenges for GPs [20,21]. Young people have reported that such hesitancy to disclose can be the result of a poor relationship with the GP or the GP’s interpersonal approaches, and fears around a loss of privacy [25], which GPs in this study acknowledged were important concerns. Furthermore, GPs were mindful of the role and impact of family support in the management of suicidal behaviour and self-harm, and how having a difficult family environment could impact upon a young person’s risk [60]. They outlined a range of strategies to facilitate rapport and encourage disclosure, such as taking a non-judgmental approach to presentations of self-harm, having a positive therapeutic relationship, and collaboratively addressing confidentiality and safety, all of which are congruent with young people’s preferences [25]. However, none explicitly discussed how they would manage or even possibly avoid involving parents or family members in the young person’s care if their involvement could potentially be harmful, and the considerations around doing so.

Regarding approaches to suicide and self-harm assessments, some GPs utilised recommended best-practice approaches, such as conversational interviewing that incorporated assessments of patient history, mental states, and protective factors [27,28,29,61]. They also generally reflected young people’s views on the importance of having a collaborative, person-centred dialogue that was guided by the young people’s needs [25]. However, due to both time constraints and feelings of discomfort and uncertainty with conducting assessments, GPs also frequently relied on “tick-box”-type screening tools to help identify the potential presence of suicide and self-harm risk factors. Conceptualising and predicting future suicidal behaviour and self-harm were identified as challenges, in particular against the backdrop of emotional and cognitive developmental factors, and concerns that suicide could occur without warning. Indeed, evidence has suggested that suicide attempts in young people can be unplanned [62,63] and that it is difficult, if not impossible, to determine the point of transition to engaging in self-harm or a suicide attempt [64,65,66]. There is also growing evidence about the inability of screening instruments and formalised risk assessments to accurately predict suicide and self-harm outcomes [67,68,69,70,71]. However, GPs in this study demonstrated conflicting and often paradoxical opinions on their value. Similar to previous studies [20,21], some GPs recognised the limitations of such tools and believed that they could overshadow the therapeutic relationship; however, others felt they could be useful as a guide or prompt to offer suggestions for inquiry and direction to management strategies, especially for those who were less experienced. Some also identified a lack of procedures in their clinics for responding to suicide and self-harm-related presentations, which has been previously reported as a barrier to effective management [72]. They felt that more standardised processes would help to alleviate the discomfort and uncertainty experienced. In considering GPs’ views and experiences alongside the best-practice evidence, we outline several practice and service suggestions below.

### 4.2. Practice and Service Implications

#### 4.2.1. Approaches to the Assessment of Suicidal Behaviour and Self-Harm

It is important to consider a number of things when exploring GPs’ approaches to suicide and self-harm assessments, and their use of and requests for tools to assist. Firstly, the needs and preferences of young people, who have reported: (1) that tick-box and screening-style approaches can be impersonal and reductionist, and reduce their experience to labels, and (2) fear that over-attention to the presence of specific risk factors at the expense of a holistic assessment may result in missed opportunities for assistance or being directed to inappropriate treatment pathways [25]. One of the key criticisms of such tools is that categorising risk into different ‘levels’ (e.g., low, medium, high) is unreliable and potentially hazardous [67,68,69,70]. Although most GPs in our study acknowledged that the use of risk assessment tools needed to be flexible and person-oriented, some wanted tools that replicated these processes. Given the overall discomfort GPs described, it is perhaps unsurprising that they still request these types of tools despite awareness of their pitfalls. Secondly, although suicidal intent is fluctuating and dynamic [73], an extensive literature base highlights key modifiable risk factors, warning signs, and protective factors for suicide and self-harm in young people [74,75,76,77]. Although it is not possible to definitively predict suicidal behaviour and self-harm, it is essential that GPs remain alert to changing situational risk factors in the young person’s life, and understand how exposure to modifiable risk factors can be associated with suicidal behaviour and self-harm. They should also recognise lesser degrees of distress as potentially serious [63], even if suicide plans are vague or non-existent. Overall, there is a need for collaborative, holistic, and skilful assessments that go beyond standardised tools and risk categorisation, and ensure that young people’s needs are met.

A number of digital tools to assist both GPs and young people have been developed. For example, there are several self-administered, pre-appointment screening tools that use adapted versions of the HEEADSSS assessment [48,49] as general psychosocial screeners for potential risk, such as the *Check Up GP* app [78], the *myAssessment* app [79], and the New Zealand-based *YouthCHAT* [80]. These aim to help the GP (or other healthcare professionals) to identify psychosocial areas of concern and then provide clinical management suggestions and referral resources. These have been shown to lead to improvements in disclosure and perceptions of patient-centred care among young people and to be acceptable to professionals [78,79,81,82]. Clinical decision support systems (CDSS) that include guidance around questions for GPs to ask, reminders of risk factors, and follow-up suggestions may also be useful for those who lack confidence. An electronic CDSS focusing on suicide risk is currently being trialled in the UK [83]. However, in line with GPs’ suggestions, such tools should be flexible and prioritise alleviating GPs’ time burdens rather than adding to them. Assessment tools should also facilitate a collaborative clinical interview-style approach, rather than feeling like a “protocol” or checklist is being administered to young people [25].

#### 4.2.2. Addressing Systemic Barriers

Systemic barriers such as unclear referral pathways, complex entry criteria for specialist services (particularly for tertiary-level services such as CAMHS), and a lack of mental health service integration into primary care align with those reported in previous studies [20,21,72,84]. Given that GPs are often the first port of call for at-risk young people, there is a clear need for solid investment to better support GP capacity in identifying and responding effectively to suicidal behaviour and self-harm. System improvements such as increased service integration, smoother referral and service pathways, improved education around service entry and access criteria, and better resourcing for integrated support from other external mental health professions (ideally, the provision of an in-house, multi-disciplinary team) may help to alleviate some of GPs’ described burdens. Past evidence suggests that keyworkers, such as community psychiatric nurses, counsellors, or psychologists, plus single points of access and enhanced service provision for self-harm in primary care, would better support GPs [18,85]. These approaches could also help to address issues around care continuity between practitioners and also may reduce engagement and follow-up problems with young people. 

#### 4.2.3. Building GP Capacity with Training

Consistent with past findings [72], the GPs in our study identified that specific suicide and self-harm training was rare, and even more so that which focused on young people. Training for GPs in identifying and responding to suicidal behaviours has long been an accepted and effective suicide prevention strategy, with training improving GP knowledge, confidence, and skills [17,86,87,88,89,90]. Furthermore, modelling studies suggest that some of the greatest reductions in Australia’s suicide rates could be achieved through suicide prevention training in primary care [19]. Given the established evidence of such training, the best-practice recommendations from GP peak professional bodies which emphasise the essential role GPs play in the management of suicide and self-harm, and the preferences of young people for GP competency in this area [17,25,28,29,61,91], the question is raised of whether suicide and self-harm specific training should be a universal requirement for GPs. 

Training could be incorporated into the core medical school curriculum [92], and extended beyond this as part of continued professional learning. It should involve meaningful and collaborative youth and GP participation, which would help to engage GPs and affirm real-world relevance. It should also include ‘active’ learning components, and be flexible in its delivery—offering both online and face-to-face formats [20]. Content could include conducting collaborative psychosocial suicide and self-harm assessments and safety planning, in addition to addressing core communication skills to better enable identification of suicide and self-harm and promote good rapport, as young people have highlighted that this is an area that needs improvement [25], and past research has demonstrated problems in the way healthcare professionals frame suicide assessment questions [93]. Although training in assessments and interventions, and communication and therapeutic skills, may be met with resistance from a small number of GPs who believe such interventions are outside their role or that communication skills are “set”, research has shown that collaborative psychosocial assessments and safety planning improves disclosure of risk and outcomes on suicide-related measures, and are acceptable and feasible to young people and practitioners [94,95,96,97,98,99,100,101]. Similarly, suicide and self-harm training that focuses on improving communication and relational skills is effective [102,103,104]. As such, training which targets these specific areas would also likely be beneficial. 

### 4.3. Strengths and Limitations

Participants self-selected to take part in the study; therefore, those who demonstrated interest may have been particularly passionate or knowledgeable on the topic of youth suicide/self-harm. Additionally, the sample was collected from a geographical catchment in Western Australia which is predominately a disadvantaged and low-socioeconomic area [36]. Thus, the sample may have limited generalisability. However, we interviewed a mix of participants who were recruited purposively from a range of clinic types to help ensure diversity in their experiences, including clinics that were not youth-specific (and thus may have less exposure to youth suicide/self-harm risk), and those that were exclusively focused on young people.

As GPs in the group interviews were colleagues from the same practices, this may have influenced their expressed opinions through self-presentation and impression management strategies or group-think effects [105,106]. However, group interviews can also stimulate deeper discussion and increase disclosure around sensitive topics and information [31]. Lastly, demographic data was missing from nearly half of the sample, who failed to complete the questionnaire and did not respond to follow-ups. Therefore, the demographic information presented may not be fully representative of the sample. 

Despite these limitations, the study had several strengths. To the best of our knowledge, it is the first qualitative study examining GPs’ perspectives and practices on managing youth suicidality and self-harm in an Australian setting. More broadly, the findings also contribute to the growing body of literature on this topic, and share similarities with GPs’ described experiences and challenges elsewhere. Methodological rigour was enhanced by the use of consensus approaches to improve analysis fidelity [43], and we reported the study in line with the COREQ guidelines [30]. Furthermore, we used a combination of individual and group interviews, which can enhance data richness and depth of understanding [32]. 

## 5. Conclusions

Although primary care is a main point of contact for young people at risk of suicidal behaviour and self-harm, this study identified a number of inter-related gaps and challenges that GPs experience. These included developing rapport, facilitating disclosure, negotiating confidentiality and family involvement, discomfort related to working with suicide and self-harm presentations, and issues with assessing a young person’s risk. They also involved a range of systemic and structural barriers, including time constraints and service referral and integration issues, in addition to perceptions and beliefs around GPs’ roles. Despite the need for improved service delivery, there is significant potential for GPs to play a key role in youth suicide prevention. Engaging in collaborative and person-centred assessments, using effective interpersonal and communication skills, and better resource provision could all help to maximise the potential of the primary care consultation. 

## Data Availability

Anonymised transcripts are available at Orygen, Suicide Prevention. These may be accessed upon reasonable request by contacting author I.B.W. at india.bellairs-walsh@orygen.org.au. The data are not publicly available due to ethical and privacy restrictions.

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
