# Peer review of "Working with Young People at Risk of Suicidal Behaviour and Self-Harm: A Qualitative Study of Australian General Practitioners’ Perspectives"

_ijerph, 2021, doi:10.3390/ijerph182412926_

Round 1

Reviewer 1 Report

Thank you for the opportunity to review this paper. Overall, it reads well and the topic is relevant, and provides a new insight into the complexities GP's have to face when working with and assessing young people for suicidal  behaviour and self-harm.  I would recommend the paper be accepted with minor corrections. The paper could be improved by the authors expanding upon the methods section. They need to  explain what type of qualitative study this was, for example rather than just saying it was a qualitative study they need to explain what theoretical approach underpinned their study.  Did they use a  phenomenological approach, or grounded theory? Could the authors also explain how they managed the differences in the researchers skill sets in relation to how the interviews were conducted, thus ensuring the validity and reliability of the study.

Reviewer 2 Report

Dear authors of the paper «Working with young people at risk of suicidal behaviour and self-harm: A qualitative study of Australian General Practitioners perspectives”

I am impressed by the initiative and importance of these findings in general and argue that the paper adds important knowledge to the fields of general practice and suicidology. However, I think that the results are of importance for more than Australian GPs, as several of the themes are relevant for larger parts of the world. E.g. European health care systems that are similarly organized. But also the themes that points out more universal phenomena in the relation with both the adolescent and the parent.

I will therefore suggest that the in the introduction section:

Line 82 : Whilst a large proportion of the existing research on GPs’ experiences comes from international sources, particularly the UK, to date Can you add a reference to this research and if possible elaborate on why there is only missing research from Australia (I think it is to modest and that there is a need only from an Australian setting and thus the research question is also relevant in an international setting )

Minor: line 67: Please elaborate or explain what’s in: resource constraints [20,21],

Methods section

Line 138-140: Depression, as a known 138 risk factor for both suicidal behaviour and self-harm [7,38], was also included in the interview question wording, however was considered outside the scope of this paper, which 140 has a specific focus on suicidal behaviour and self-harm.

I partly understand this consideration, but depression is an important part of the suicidal process. In our interview study of GPs and suicidality among adults, many argued that they often discovered and made an agreement with their patients that they were to some degree depressed before they address the question about suicide thoughts (they often used the Montgomery and Asberg depression scale where suicidality was the last question). Even if you argue that depression is out of scope for this paper, maybe that should be described more thoroughly?

Discussion section

Line 570-571 Dealing with hesitancy from young people to disclose suicidality and self-harm, as well as balancing family attendance at appointments

The involvement of parents is an important finding and I think that the paper would improve by discussing this further. E.g. in some cases the parenting style and adversity in the home might be associated with the suicidality and self-harming behaviour and thus the adolescent might be reluctant to disclosure. Or in other cases they need to be informed.

Although probably also out of scope for this paper, It would have been interesting to know a bit more about the interventions carried out in general practice with no need for specialist health care and maybe whether the GPs had any thoughts about their preventive role in hindering enough suicidal behaviour?
